# VibeSpace: Automatic vector embedding creation for arbitrary domains and mapping between them using large language models

## Abstract

We present VibeSpace; a method for the fully unsupervised construction of interpretable embedding spaces applicable to arbitrary domain areas. By leveraging knowledge contained within large language models, our method automates otherwise costly data acquisition processes and assesses the similarity of entities, allowing for meaningful and interpretable positioning within vector spaces. Our approach is also capable of learning intelligent mappings between vector space representations of non-overlapping domains, allowing for a novel form of cross-domain similarity analysis. First, we demonstrate that our data collection methodology yields comprehensive and rich datasets across multiple domains, including songs, books, and movies. Second, we show that our method yields single-domain embedding spaces which are separable by various domain specific features. These representations provide a solid foundation upon which we can develop classifiers and initialise recommender systems, demonstrating our methods utility as a data-free solution to the cold-start problem. Further, these spaces can be interactively queried to obtain semantic information about different regions in embedding spaces. Lastly, we argue that by exploiting the unique capabilities of current state-of-the-art large language models, we produce cross-domain mappings which capture contextual relationships between heterogeneous entities which may not be attainable through traditional methods. The presented method facilitates the creation of embedding spaces of any domain which circumvents the need for collection and calibration of sensitive user data, as well as providing deeper insights and better interpretations of multi-domain data.

## 1 Introduction

> *"In a universe of endless variation, each entity is a unique symphony of vibes—each different, each iridescent in its own irreplaceable way, harmonizing in the grand composition of existence."* - ChatGPT4

In recent years, embedding spaces have played a pivotal role in transforming raw data into interpretable and quantifiable vectors, especially in domains that require semantic information extraction and similarity assessment. Traditionally, constructing such spaces requires substantial datasets - the generation of which can involve costly data gathering and processing. This requirement for substantial datasets to initialise embedding-based systems in data-scarce environments defines the cold-start problem, a frequent hurdle in initiating systems like classifiers or recommenders. As a solution to the cold-start problem, we introduce 'VibeSpace', a novel approach for fully automated synthetic data collection and unsupervised creation of both single-domain embedding spaces and mapping functions between these spaces. Further, we present methods for interactively querying generated vector spaces, a process which brings insight and interpretability to what might otherwise be an opaque technique.

For many embedding-based systems to function well, it is required that vector embeddings place "similar" items in proximate locations, ensuring that such spaces are contextually apt. In the context of recommender systems, "similarity" can be a nebulous term, merely meaning whatever human users would consider to be "similar". So, how can we ascertain item similarity without an

exhaustive human evaluation? We argue that state-of-the-art large language models (LLMs) offer a promising solution; given their extensive training on repositories of human-authored text, these models should have some knowledge of what would be perceived as similar, particularly within a single domain (e.g. films which should be considered similar to each other). Notably, however, many state-of-the-art LLMs have knowledge of what humans would consider similar that can bridge the gap between disparate domains and have the ability to draw associations between seemingly unrelated entities. When prompted, for example, state-of-the-art LLMs will argue that Monopoly is more similar to the City of London than Santa Monica Beach, identifying underlying "vibe-based" similarities that would be difficult to capture with standard data-gathering techniques. Given these capabilities, LLMs are an invaluable tool for the construction of vector spaces for both single and multiple domains.

Our paper also introduces the utilisation of LLMs as autonomous agents dedicated to data collection. These LLMs serve as knowledge repositories spanning innumerable domains which can be seamlessly accessed by querying, offering an immediate route to data acquisition. Unlike conventional data-gathering methods, utilising LLMs presents a more straightforward and economically viable alternative. Further, LLMs can be easily tailored or fine-tuned to particular specialised domains. Thus, our approach could be used to automatically generate datasets for domains where even current state-of-the-art LLMs have little knowledge.

We identify two primary avenues of value stemming from our work. Firstly, our methodology holds immense potential in generating datasets across any domain. This universal applicability could revolutionise data-driven fields, providing them with rich, contextually relevant datasets that were previously difficult or expensive to obtain. Secondly, our approach addresses a long-standing challenge within the realm of recommender systems: the new-community cold-start problem. By facilitating the creation of vector embeddings that can be created in the absence of data, we allow for easily initialised systems that can be refined if desired as user data accumulates.

## 2 Prior Work

### 2.1 Word Embeddings

In Natural Language Processing and Information Retrieval, capturing the essence of words for computational tasks is crucial. Word embeddings offer a solution by converting words into vector representations. The spatial distances between these vectors reflect their semantic relationships Li & Yang (2018). Embeddings of this kind were first introduced as a tool for Information Retrieval systems by Salton et al. (1975), who represented both queries and documents in a vector space, with each dimension tied to a specific vocabulary term. This approach has significantly influenced subsequent advancements in understanding linguistic context and semantics in computational models Pennington et al. (2014).

A significant contribution in this realm comes from Turian et al. (2010). Their work considers word representation as a semi-supervised problem, utilizing the context in which a word appears within a text to infer its semantic meaning - in other words, "You Shall Know a Word by the Company It Keeps" Widdowson (2007).

Subsequent to this, Mikolov et al. (2013a) introduced two novel architectures: Continuous Bag of Words (CBOW) and Skipgram. These architectures were foundational to the development of the Word2Vec (Mikolov et al. (2013a;b;c)) models, which have since become a benchmark in the field of word embeddings. The Word2Vec approach, particularly through these architectures, efficiently captures syntactic and semantic word relationships, revolutionising subsequent models and methods in the domain Levy & Goldberg (2014); Goldberg (2016).

### 2.2 Recommendation Systems

Companies continually confront the issue of effectively presenting users with relevant items from their inventories. To mitigate this challenge, recommender systems, rooted in both collaborative and content-based filtering techniques, have been developed Ricci et al. (2010). These systems tailor selections based on user preferences and historical data Adomavicius & Tuzhilin (2005). They find applications in a broad spectrum of domains. While their adoption in sectors like entertainment, for

recommending books or movies, is widely acknowledged Bobadilla et al. (2013), they've also made significant inroads into more specialized areas. This includes making recommendations in financial services, suggesting telecommunication equipment, and even aiding in software system selections Zhang et al. (2019); Felfernig et al. (2021).

Recommender systems typically employ four strategies for recommendation selection. The content-based recommendation approach is the first, wherein systems suggest items to a user based on their similarity to items the user has previously engaged with Pazzani & Billsus (1997). Second is the collaborative filtering recommendation systems, where items are recommended to a user based on positive ratings from users with similar preferences Konstan et al. (1997). Lastly, the knowledge-based recommendation approach compares a given user requests with item features to infer the most relevant suggestions Burke (2000).

Recommendation systems each come with their strengths and challenges. A significant issue they face is the cold start problem, which surfaces when a system encounters a new item or a new user, making it tough to provide relevant recommendations due to insufficient initial ratings Wahab et al. (2022). This cold start problem manifests in three primary scenarios: the "new community", "new item", and "new user" problems Bobadilla et al. (2012).

In content-based recommendation systems, the "new user" cold start problem is prevalent; when an individual user with no prior interactions is added, the system, which primarily relies on historical user ratings, finds it hard to generate accurate recommendations Lika et al. (2014). Collaborative filtering systems, on the other hand, face the "new item" cold start problem more acutely. When the system introduces a new item that hasn't received any ratings, it remains largely overlooked in the recommendation process. Similarly, with a new user who hasn't provided any ratings, determining their preferences becomes a hurdle, leading to less accurate recommendations Natarajan et al. (2020).

The "new community" cold start problem is particularly challenging, as it relates to a completely new set of users without any established interaction history. This lack of data makes it difficult to assess similarity of items in the catalogue, gauge preferences and offer relevant recommendations. Though a significant amount of work has been undertaken to tackle the "new item" and "new user" cold start problems, relatively little research has examined how to tackle the final of these problems Gope & Jain (2017); Sahebi & Cohen (2011).

Multi-domain recommenders are systems designed to make suggestions across a variety of unrelated domains, for instance, recommending both movies and restaurants to a user based on their preferences. The complexity of these systems significantly increases compared to single-domain recommenders due to several challenges. First, they must address the inherent heterogeneity of data across domains, where different types of interactions, user behaviors, and item attributes need to be harmonized for consistent recommendation Fernández-Tobías & Cantador (2014). Moreover, sparsity becomes an acute issue; while a user might have many interactions in one domain (e.g. book reviews), they might have very few in another (e.g. travel experiences) Cantador et al. (2015). Additionally, capturing the transferable knowledge between unrelated domains without diluting domain-specific nuances is a non-trivial task Maake et al. (2019). Ensuring that the recommendations are both contextually and semantically relevant across all domains remains a challenge in the development of effective multi-domain recommenders.

## 2.3 LARGE LANGUAGE MODELS

Recent years have seen major advancements in language modeling. Language models aim to predict the likelihood of word sequences, helping predict future or missing words. This area has gained much attention, as outlined in works like Zhao et al. (2023). A significant step forward in this field has been the development of Large Language Models (LLMs). These LLMs excel in various tasks such as question answering, information retrieval, and translation Brown et al. (2020); Devlin et al. (2018).

One of the standout features of LLMs is their ability to produce context-aware word embeddings. Gao et al. (2021) described two methods, supervised and unsupervised, to generate these embeddings from LLMs. Taking the unsupervised approach a step further, Chuang et al. (2022) introduced

a technique where the model differentiates between original and slightly altered sentences to create more refined embeddings.

Due to these breakthroughs, the potential uses of LLMs have increased dramatically. Currently, LLMs are being considered for use in education Baidoo-Anu & Ansah (2023), public health Biswas (2023b), as an aid for climate research Biswas (2023a), and many more.

## 3 Methods

### 3.1 Data Collection

Here we describe our automated data collection methodology. We first queried ChatGPT3.5 Brown et al. (2020) using a strategy designed to obtain maximum information about entities within a given domain, and the similarity information between those entities. This data is used to create the single domain embedding spaces as described in section 3.1.1. Second, we queried ChatGPT3.5 using a strategy designed to get information about semantic relationships between entities in distinct non-overlapping domains. This data was used to create the multi-domain mappings as described in section 3.1.2. We chose to focus on three non-overlapping domains for this study; movies, books, and songs. These domains were selected due to the availability of comprehensive reference data, meaning entities in our vector spaces could be verified against real-world meta-information. Data was accessed for movies using the IMDb API, for songs using the Spotify API, and for books using the Google Books API combined with the Goodreads book-graph datasets, Wan & McAuley (2018) Wan et al. (2019).

### 3.1.1 Vector Space Creation Data

---

**Algorithm 1** Auto Single Domain Data Collection

---

**Require:** Domain name: $D$, number of lists to generate: $N$, initial domain pool size: $n_0$, number of similar items to give per list: $n_1$, formatting instruction prompt (optional): $f$, similarity definition prompt $s$.
1: Deline $L$ = []
2: Query LLM: "Give me a list of $n_0$ $D$s." - Add $f$ if given.
3: Parse returned string into list $E$
4: **repeat**
5:     Define $E_{new}$ = []
6:     **for** entity $e$ in $E$ **do**
7:         Query LLM: "Give me a list of $n_1$ $D$s most similar to '$e$'. The list should be in similarity order." - Add $f$ and $s$ if given.
8:         Parse returned string into list $l$, add $e$ as first element of $l$
9:         Add unique elements of list $l$ to $E_{new}$
10:     **end for**
11:     Set $E = E_{new}$
12:     Shuffle $E_{new}$
13: **until** length($L$) $\geq N$
14: Return $L$

---

To function with our the modified Word2Vec algorithm, we require lists of 'sentences'. In our case, these sentences should consist of tokens representing similar entities of the same domain. For example, when collecting data to generate a sensible film embedding space, we require many sentences which are each lists of similar films.

Our system uses a large language model as a data collection agent and similarity judge. After inputting a given domain e.g. 'song', the prompt 'Generate me a list of 50 songs' will be sent to ChatGPT3.5, which will return $D = [d_1, d_2, ..., d_{50}]$ Iterating through this list, we ask our LLM to generate us a list of 10 domain objects most similar to each of the entities in our current dataset, in similarity order. For example 'Give me a list of 10 songs most similar to *Miami by Will Smith*'. If these entities are novel, they are appended to our $D$ entity list. This process is repeated for each domain of interest until a fixed number of sentences have been generated.

In our experimental setup, data was generated across the three selected domains. For both the book and song domains, we produced 25000 lists of similar domain entities. However, we generated extra data for the movie domain, which totalled 50000 lists. This intentional disparity in data generation for the movie domain was driven by our objective to benchmark our automatically constructed dataset with the established MovieLens dataset Harper & Konstan (2015).

### 3.1.2 VECTOR SPACE MAPPING DATA

---

**Algorithm 2** Auto Cross Domain Data Collection

---

**Require:** Domain name 1: $d_1$, Domain name 2: $d_2$, list of known entities within $d_1$: $D_1$, list of known entities within $d_2$: $D_2$, number of similarity lists to generate: $N$, number of similar items to give per list: $n$, formatting instruction prompt (optional): $f$, similarity definition prompt $s$.
1: Define list of training pairs $M$ = []
2: **repeat**
3:     **for** random entity $e_1$ in $D_1$ **do**
4:         Query LLM: "Give me a list of $n$ $d_2$s most similar to the $d_1$ '$e$'." - Add $f$ and $s$ if given.
5:         Parse returned string into list $l$.
6:         Set $l'$ = elements of $l$ which exist in $D_2$
7:         Add $(e_1, l')$ to $M$
8:     **end for**
9: **until** lenth$(M) \geq N$
10: Return $M$

---

To generate our data for generating mapping functions between vector spaces of two domain, we require cross-domain training pairs. Our system again uses a large language model as a data collection agent and similarity judge. We first randomly sample an entity from our first domain dataset $D_1$, then prompt our LLM to return a list of 5 entities in our second domain. If any of the returned entities exist within our dataset $D_2$, we add both entities as a training pair into the dataset.

We sampled 10000 elements of each domain and generated their corresponding training pairs for both other domains. E.g. 10000 elements in the film domain are mapped to points in the book domain.

### 3.2 ENTITY2VEC: BUILDING VECTOR SPACE

We have adapted the Word2Vec approach, as described by Mikolov et al. (2013a;b;c) , to construct 100-dimensional vector spaces based on the lists of similar domain entities constructed in section 3.1.1. Initially, we augment our dataset by permuting and resampling entities from our similarity lists. Entities that are more alike appear in lists together more frequently in this augmented dataset, reflecting their positions in the similarity list.

We then approach this reshuffled list of entities as if it were a textual corpus for a conventional Word2Vec procedure. Here, each entity is treated as a unique token in the classical implementation would be. Importantly, in our adapted Word2Vec method, the actual names or textual content of the entities play no role in their final positions within the vector space. Instead, their positions are solely determined by their placements within the similarity lists.

### 3.3 VIBEMAP: MAPPING BETWEEN VECTOR SPACES

We employ a straightforward autoencoder architecture to execute domain mapping. This autoencoder is trained on the vector representations of the training pairs delineated in section 3.1.2. To enhance the robustness of our model, we introduce Gaussian noise to our training pair vectors during the training process.

## 4 RESULTS

### 4.1 ACCURACY & COMPLETENESS OF DATASETS

We first examine the quality of our generated datasets. Using the Last.FM API, Google Books API, and IMDB API, we examine the fidelity of our song, book, and movie datasets, respectively. We are also able to quantify the number of unique entities in each of our domains. These results are displayed in Table 1.

|              | Movies | Songs  | Book   |
| ------------ | ------ | ------ | ------ |
| Total Unique | 30404  | 30579  | 52600  |
| Verified     | 83.4%  | 95.46% | 99.97% |
| Invented     | 2.0%   | 4.54%  | 0.03%  |
| Non-Domain   | 14.6%  | 0%     | 0%     |

Table 1: Accuracy table for generated datasets in three domains' movies, songs, books.

We first note that accuracy varies significantly by domain, from 99.97% accuracy for the books domain, to 83.4% for movies. Second, we note that the high number of non-domain entries in the movie dataset is due primarily to television shows being returned by the language model in place of films. As language models improve, as will the fidelity of datasets generated using this method. Lastly, we again note that these datasets were generated completely automatically, simply by entering the name of the domain of interest.

We next compare our movie dataset to the MovieLens dataset; a standard benchmarking dataset for movie recommendations. Within the MovieLens dataset, there are 27279 unique movie titles. Of these titles, the dataset generated using our method contains 17833, or 65.4% coverage. However, our dataset also contains 12571 titles which are not in the MovieLens dataset, highlighting our methods ability to uncover niche entities by following chains of "similar entities" until we arrive at the obscure.

### 4.2 GENERATED VECTOR SPACES

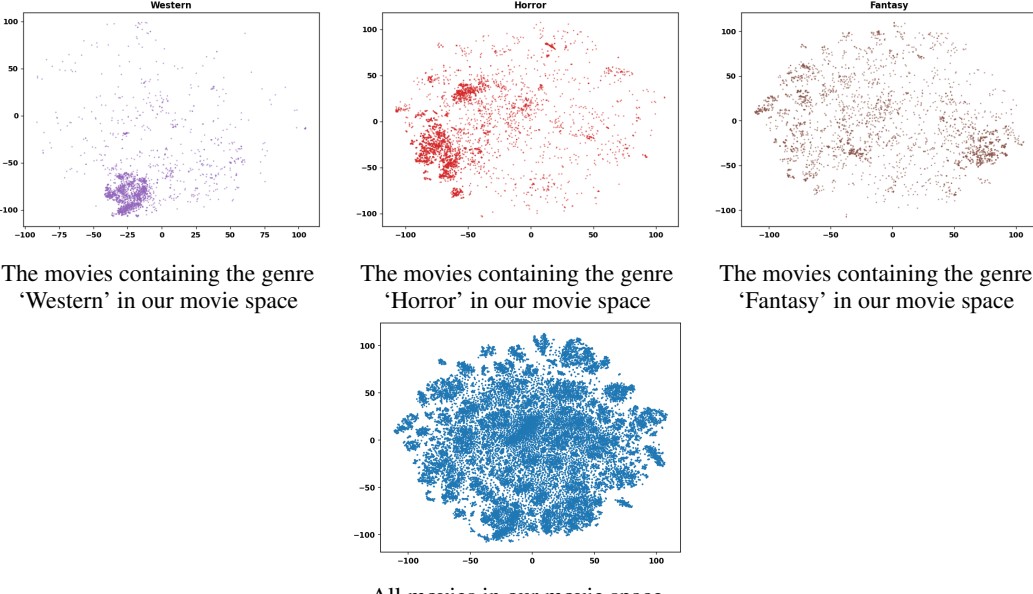

| The movies containing the genre 'Western' in our movie space | The movies containing the genre 'Horror' in our movie space | The movies containing the genre 'Fantasy' in our movie space |

All movies in our movie space

Figure 1: TSNE projections of the movie space (bottom), as well as the positions of various genres within that space (top).

Employing the technique described in section 3.2, we constructed vector embeddings for our three designated domains: movies, books, and songs. To evaluate these domains, we assessed their sepa-

rability based on features accessible through the respective APIs. Genre was selected as the feature for this separability test, given its relevance across all our chosen domains. TSNE projections of one of these spaces, as well as the positions of various genres within that space, are shown in Figure 1.

| Genre | Action | Adventure | Animation | Biography | Comedy | Crime | Documentary | Drama | Family | Fantasy | Film-Noir |
|---|---|---|---|---|---|---|---|---|---|---|---|
| Acc | 0.879 | 0.902 | 0.973 | 0.943 | 0.814 | 0.871 | 0.922 | 0.729 | 0.939 | 0.927 | 0.991 |
| F1 | 0.641 | 0.537 | 0.663 | 0.281 | 0.638 | 0.488 | 0.659 | 0.716 | 0.599 | 0.430 | 0.661 |

| Genre | History | Horror | Music | Musical | Mystery | Romance | Sci-Fi | Sport | Thriller | War | Western |
|---|---|---|---|---|---|---|---|---|---|---|---|
| Acc | 0.952 | 0.939 | 0.950 | 0.957 | 0.914 | 0.840 | 0.948 | 0.977 | 0.855 | 0.969 | 0.976 |
| F1 | 0.288 | 0.691 | 0.515 | 0.398 | 0.385 | 0.561 | 0.624 | 0.558 | 0.597 | 0.488 | 0.785 |

Table 2: Accuracy & F1 table for movie entity classifications; genres generated by IMDB API.

| Genre | 60s | 70s | 80s | 90s | Alternative | Alternative Rock | Female Vocalist | Classic Rock | Country | Dance | Electronic | Singer-Songwriter |
|---|---|---|---|---|---|---|---|---|---|---|---|---|
| Acc | 0.969 | 0.974 | 0.967 | 0.976 | 0.963 | 0.974 | 0.943 | 0.957 | 0.967 | 0.962 | 0.955 | 0.981 |
| F1 | 0.515 | 0.614 | 0.639 | 0.488 | 0.476 | 0.555 | 0.688 | 0.678 | 0.771 | 0.436 | 0.535 | 0.414 |

| Genre | Blues | Folk | Hip Hop | Indie | Jazz | Latin | Oldies | Pop | Rap | RnB | Rock | Soul |
|---|---|---|---|---|---|---|---|---|---|---|---|---|
| Acc | 0.985 | 0.975 | 0.983 | 0.972 | 0.972 | 0.978 | 0.973 | 0.901 | 0.945 | 0.976 | 0.904 | 0.968 |
| F1 | 0.688 | 0.568 | 0.395 | 0.623 | 0.626 | 0.547 | 0.556 | 0.587 | 0.407 | 0.619 | 0.647 | 0.538 |

Table 3: Accuracy & F1 table for song entity classifications; genres generated by Last.FM API.

| Genre | Art | (Auto)Biography | Body, Mind & Spirit | Business & Economics | Graphical Novels | Cooking | Fiction | History | Juvenile Fiction | Juvenile Non-Fiction | Literary Criticism |
|---|---|---|---|---|---|---|---|---|---|---|---|
| Acc | 0.993 | 0.951 | 0.986 | 0.980 | 0.986 | 0.995 | 0.841 | 0.917 | 0.956 | 0.989 | 0.966 |
| F1 | 0.00 | 0.224 | 0.500 | 0.481 | 0.647 | 0.722 | 0.646 | 0.377 | 0.478 | 0.281 | 0.00 |

| Genre | Nature | Performing Arts | Philosophy | Poetry | Political Science | Psychology | Religion | Science | Self-Help | Social Science | Young Adult Fiction |
|---|---|---|---|---|---|---|---|---|---|---|---|
| Acc | 0.986 | 0.983 | 0.979 | 0.987 | 0.982 | 0.990 | 0.933 | 0.984 | 0.991 | 0.979 | 0.976 |
| F1 | 0.240 | 0.340 | 0.400 | 0.353 | 0.00 | 0.358 | 0.613 | 0.369 | 0.00 | 0.00 | 0.619 |

Table 4: Accuracy & F1 table for book entity classifications; genres generated by Google Books API.

To measure separability of the classes, we create a 1-vs-rest classifier for each of the respective domains. We chose this over a single classifier due to significant imbalances between classes. For classification we use gradient boosted decision trees, trained using the Lightgbm package Ke et al. (2017). By requiring only a linear classifier to separate the classes to a significant degree, we show that the similar entities provided by the language model are meaningful and consistent. The full results for movies, songs and books are shown in Table 2,Table 3 and Table 4, respectively.

### 4.2.1 Querying Embedding Position Meanings in Vector Space

Another benefit of using LLMs with vector embeddings is that these embeddings can be queried using the LLM. In standard unsupervised embedding approaches, the semantic meanings associated with the positions inside the vector space can be difficult to quantify without a thorough analysis, meaning these spaces are not as interpretable as they could be. With our approach, however, it is easy to quantify and label vector positions using a single API call as demonstrated in Algorithm 3 below. Using this approach, we are able to generate easy to understand labels on the high dimensional vector space, or on the low dimensional TSNE representation of the vector space, as show in Figure 2.

---

**Algorithm 3** Auto Cross Domain Data Collection

---

**Require:** Position in vector space: $p$, Neighborhood distance: $\delta$
1: Adjectives = []
2: **repeat**
3:     Sample 10 entities $e = [e_1, ..., e_{10}]$ within $\delta$ from $p$.
4:     Sample 10 entities $e' = [e'_1, ..., e'_{10}]$ at least $\delta$ away from $p$.
5:     Query LLM: "Give me an adjective that is true for the entities $e$ but not true for the entities in $e'$". Add the returned adjective to Adjectives.
6: **until** lenth(Adjectives) $\geq$ 10
7: Return Adjectives

---

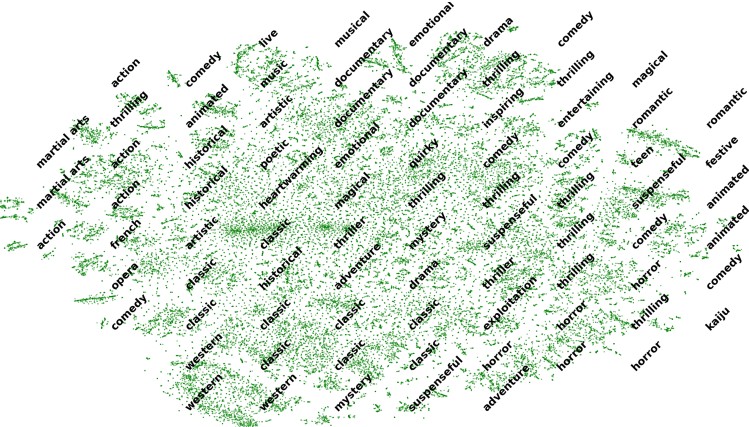

Figure 2: The movie domain vector space, with overlayed position query labels. We note identical labels being placed in nearby locations, indicating localized clusters containing primarily one single genre.

### 4.3 Consistency of Mappings

Using the method outlined in section 3.3, we generated embedding maps that bridge our three chosen domains: movies, books, and songs. Assessing the quality of these mappings is challenging due to the inherently subjective nature of cross-domain similarities. To address this, we devised two evaluation metrics tailored for our mapping functions. The first metric, symmetry error, measures the deviation from the starting point when transitioning from domain $D_1$ to $D_2$ and back to $D_1$. The second metric, triangularity error, gauges the displacement from the origin when transitioning through domains $D_1$, $D_2$, and $D_3$ in sequence, ultimately returning to $D_1$. Visualisations of these metrics are shown in Figure 3.

Using these metrics, we found that all our mapping functions achieved a very low average symmetry error when compared to a purely change based mapping, obtaining an average of 8.4% the symmetry error of a purely

random mapping. The lowest symmetry error coming when mapping between books and movies. Triangularity error was higher, but still performed significantly better than random for all orders of domain space traversal. We believe this is an encouraging result as it indicates that the mappings preserve some contextual information across domains, even with our highly simple autoencoder architecture.

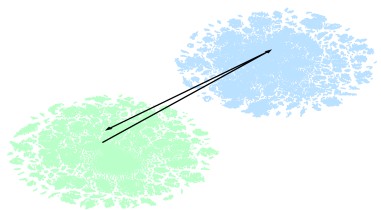 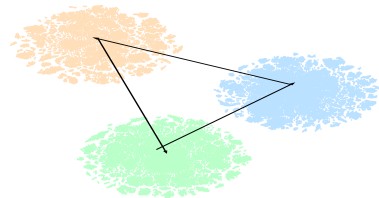

A visualization of the symmetry error metric          A visualization of the triangularity error metric

Figure 3: Visualizations of the triangularity and symmetry error metrics used to quantify quality of our cross domain mappings.

## 5 CONCLUSION

Here we have introduced and expounded on VibeSpace, an effective methodology for the unsupervised generation of interpretable embedding spaces applicable across diverse domains. The inherent strength of VibeSpace lies in harnessing the knowledge encapsulated within state-of-the-art large language models to capture important relationships between entities in generated data. This removes the need for costly data acquisition, reliance on user data, and extensive feature engineering that is common with traditional techniques. We demonstrate that our approach of simple association-based prompting is sufficient to generate high quality synthetic datasets for movie, song and book domains by validating against real-world data. Further, we show that our novel use of a modified word2vec algorithm for learning entity associations yields meaningful embedding spaces which are separable by genre, a key feature for our target domains. Finally, we introduce two novel metrics to evaluate our method for producing cross-domain mappings between our embedding spaces, symmetry error and triangularity error.

The relatively high fidelity and completeness of the generated datasets make them highly suitable for all manor of machine learning task (any form of classification task for example), but we feel our approach could have distinct value as a solution to the "new community" cold start problem for recommender systems - a quick and data free way to initialize an embedding space which could be iteratively improved upon as user data is gathered. An added advantage of our method is the interactivity component. By interactively querying the embedding space using an LLM, we can obtain useful semantic descriptors of the locations within the space. This adds a new level of interpretability and understanding to embedding based approaches.

It is worth noting here that there are several current limitations with this method. The quality and validity of generated entities is subject to the limitations of the source LLM and it's training data. It must be assumed that any harmful biases in the LLM may be preserved in the generated embedding spaces. Further, whilst we observed that hallucinated non-existent entities comprised only a small fraction of generated data for books, films and movies, this may pose a greater issue for less popular domains. However, it is equally true that as these language models improve, so will the efficacy of our presented method

For future research directions, a more exhaustive evaluation of the coherency of cross-domain mappings generated by VibeSpace should be undertaken. This would involve assessing the semantic integrity and relevance of the relationships established between different domains, and building an interactive query strategy similar to that presented here for single domains, to understand the reasoning behind mapped connections. Rigorous evaluation mechanisms, potentially involving domain experts and user feedback, could provide valuable insights into the robustness and applicability of these mappings in real-world scenarios.

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
