# OpenReview forum: "VibeSpace: Automatic vector embedding creation for arbitrary domains and mapping between them using large language models"
_ICLR.cc/2024/Conference — Submitted to ICLR 2024_

### Official Review · Reviewer_zjbm · 2023-10-30

**Soundness:** 1 poor
**Presentation:** 2 fair
**Contribution:** 1 poor
**Rating:** 1
**Confidence:** 5

**Summary:**

This paper presents a method of prompt engineering for automatic vector embedding creation via LLM.
However, this method is oversimple and there is lack of solid experiments to test the methods.

We suggest the authors:
1. Some incremental work cannot be present in top conference. There shall be insights into problems for top papers.
2. Address more your work because the background that we all know shall not be present more than one sentence.
3. Various experiments are necessary top access your work. You can enhance your experiment design.

**Strengths:**

No

**Weaknesses:**

An oversimple model presenting no insight.

**Questions:**

No

---

### Official Review · Reviewer_irAu · 2023-10-31

**Soundness:** 2 fair
**Presentation:** 2 fair
**Contribution:** 2 fair
**Rating:** 5
**Confidence:** 4

**Summary:**

This paper presents VibeSpace, a technique that utilizes large language models to automatically generate vector embedding spaces for any domain in an unsupervised way. VibeSpace enables the evaluation of entity similarity and the creation of intelligent mappings between vector spaces of non-overlapping domains. The study includes experimental results that assess the quality of the generated datasets, the interpretability of the embedding spaces and the production of cross-domain mapping based on LLM.

**Strengths:**

The data acquisition process is fully automated with LLMs, achieving good quality as validated in Section 4.1.

**Weaknesses:**

1. Section 3.3 lacks clarity. How the mappings are constructed is not clearly elaborated
2. Whether the embeddings generated by VibeSpace can really help the cold-start problem in recommender system as claimed in the paper remains to be validated.
3. Section 4.1 did a comparison of the proposed LLM-based dataset with non-LLM-based one, while Section 4.2 doesn't. How much would the embedding spaces built upon the LLM-based be better than non-LLM based ones?

**Questions:**

1. In Figure 2, we can see that the points with the same label (e.g. thrilling) actually scatters around with other labels stepped in (instead of concentrating in one area). Does this mean that the embedding spaces are not quite ideal to support the claim of "meaningful and interpretable positioning within vector spaces"?

---

### Official Review · Reviewer_jsQn · 2023-10-31

**Soundness:** 2 fair
**Presentation:** 2 fair
**Contribution:** 1 poor
**Rating:** 3
**Confidence:** 4

**Summary:**

The paper, titled "VibeSpace: Automatic Vector Embeddings for Multimedia Data", presents a novel approach to automatically generate vector embeddings for multi-domain data. The authors claimed VibeSpace can collect or extract extensive relevant datasets for an arbitrary domain, and it can address the cold-start problem in recommender systems.

VibeSpace first collect relevant datasets in terms of sentences consisting of lists of similar entities of the same domain. And then use Word2Vec to learn entity embeddings. Finally, it utilized autoencoder to perform domain mapping.

**Strengths:**

1. The idea of collecting relevant or similar entities to enrich the context is intuitive.
2. The author provided an actionable pipeline to extract embeddings based on the access to the knowledge rich LLM.
3. The author provided visualizations for the embedding space that VibeSpace provided which help the audience to understand.

**Weaknesses:**

1. Lack of novelty, the idea of extracting similar text/entity to enrich the context has been studied with [1]. The example reference extracts data from a given corpus and this work extract from LLM.
2. Unclear description about Sec 3.3. A lot of details are missing with respect to vector space mapping.
3. Lack of comparison in the design. The author uses Word2Vec to learn the embedding. However, there is no discussion on why this method is chosen and if there are any alternatives and why Word2Vec is superior to its competitors.
4. Datasets chosen does not consist with the claim. The author claims that in Sec 3.1 movies, books, and songs are non-overlapping domains, which is not convincing. For example, a lot of movies are originated from books.
5. Unclear metrics in Sec 4.2. In Table 2-4, a lot of figures are presented but without further explanation.

[1] Meng, Yu, et al. "Weakly-supervised neural text classification." proceedings of the 27th ACM International Conference on information and knowledge management. 2018.

**Questions:**

1. Why do you use Word2Vec, are there any alternatives?
2. What did you do for Sec 3.3? More details are urgently needed in this section.
3. What is your purpose by putting Figure 1 here? What are those figures for?
4. For Table 2-4, how can we interperate those tables? How to measure the quality of the separation? Some of the F1 scores are 0.00.
5. One of your contributions is about cold-start problem. Is there any empirical result that can support your claim?

---

### Official Review · Reviewer_8jfG · 2023-11-04

**Soundness:** 2 fair
**Presentation:** 3 good
**Contribution:** 2 fair
**Rating:** 3
**Confidence:** 4

**Summary:**

The paper presents VibeSpace, which extracts entities and relations of entities (mainly in the domains of movies, songs, and books), both in-domain and cross-domain. The author(s) then trains a variant of Word2Vec to get the synthetic vector space. Experiments on label classification, visualization, and mapping consistency are conducted.

**Strengths:**

1. The idea is interesting, i.e., synthesizing vector space via prompting LLMs to generate entities and relations of entities, especially those cross-domain relations.
2. Timely study on how well can LLMs generate entities and relations of entities.
3. The paper is well-written and easy to follow.

**Weaknesses:**

1. Over-claiming.
    1. In the Abstract, the author(s) said "These representations provide a solid foundation upon which we can develop ... and initialise recommender systems, demonstrating our methods utility as a data-free solution to the cold-start problem".
    2. In the Instruction, the author(s) said that "our approach addresses a long-standing challenge within the realm of recommender systems".
    3. However, no experiment is conducted on how well the proposed vector space actually alleviates cold-start problem. As the author(s) said in Conclusion: "we feel our approach could have distinct value as a solution to the cold-start problem for recommender systems", the statements in Abstract and Introduction are more like what the author(s) will do in their future work, misleading the audience about how far the state has reached.
2. The author(s) said that "This removes the need for costly data acquisition". However, compared to traditional data engineering, whether calling large language models hundreds of thousands of times really makes the cost lower is a problem.
3. The paper only examines one large language model, i.e., via calling ChatGPT API. It could be better if more LLMs like LLaMA, Vicuna are evaluated.
4. Experiments in Figure 3 show that the proposed VibeSpace has good mapping consistency. However, we still do not know how existing methods perform on the consistency of mapping. It could be better if experiments that compare VibeSpace with other baselines were conducted.
5. Code, datasets, or even samples of extracted entities, are not available.
6. "our dataset also contains 12571 titles which are not in the MovieLens dataset, highlighting our methods ability to uncover niche entities by following chains of 'similar entities' until we arrive at the obscure". In my opinion, the statement is biased because MovieLens dataset (e.g., depending on the used version, if is MovieLens-1M, the movies are all before 2003) can be not so up-to-date compared to LLMs (e.g., 2019 for ChatGPT).

**Questions:**

There has been evidences that LLMs tend to generate popular entities. Whether the proposed method will suffer from popularity bias?

---

### Meta-Review · Area_Chair_Xjgk · 2023-12-05

**Metareview:**

This paper proposes to establish a synthetic vector space for entities and relations  using an LLM, which is quite interesting. The vision of leveraging such technology to combat the cold-start problem in recommender systems is especially notable; however, all reviewers have expressed several concerns about the current state of the paper that must be addressed.

**Strengths:**

1. **Original**: The proposal to synthesize vector space via prompting LLMs is unique and an interesting direction for research. Exploring the potential of LLMs in the context of entity and relation generation is timely and relevant to current interests across various domains.

2. **Clarity**: The overall presentation of the paper is good, making it accessible to a readership interested in this work's intersection of NLP and recommender systems.

**Weaknesses:**

1. **Over-Claiming and Lack of Evidence**: Reviewers unanimously agree that the paper makes assertions in the abstract and introduction about solving the cold-start problem without providing empirical evidence.

2. The approach, particularly in Section 3.3, is not detailed enough to fully capture the architectural and technical nuances necessary for academic rigor.

3. There is a noticeable absence of comparison with existing benchmarks or methods that significantly impedes the assessment of VibeSpace's effectiveness and novelty. The paper lacks a comprehensive discussion on the metrics used and their relevance to the study's goals.

**Justification For Why Not Higher Score:**

All reviewer consistently vote for rejection of this paper and authors do not give response to the reviews raised by reviewers.

**Justification For Why Not Lower Score:**

N/A

---

### Decision · Program_Chairs · 2024-01-16

Reject